# Epstein Barr Virus Exploits Genetic Susceptibility to Increase Multiple Sclerosis Risk

**DOI:** 10.3390/microorganisms9112191

**Published:** 2021-10-21

**Authors:** Fabienne Läderach, Christian Münz

**Affiliations:** Viral Immunobiology, Institute of Experimental Immunology, University of Zurich, 8057 Zurich, Switzerland; laederach@immunology.uzh.ch or

**Keywords:** HLA-DRB1*1501, EBNA1, CD4^+^ T cells, antigen-presenting cell (APC), CD20, humanized mice, lymphoblastoid cell line (LCL)

## Abstract

Multiple sclerosis (MS) is an autoimmune disease of the central nervous system (CNS) for which both genetic and environmental risk factors have been identified. The strongest synergy among them exists between the MHC class II haplotype and infection with the Epstein Barr virus (EBV), especially symptomatic primary EBV infection (infectious mononucleosis) and elevated EBV-specific antibodies. In this review, we will summarize the epidemiological evidence that EBV infection is a prerequisite for MS development, describe altered EBV specific immune responses in MS patients, and speculate about possible pathogenic mechanisms for the synergy between EBV infection and the MS-associated MHC class II haplotype. We will also discuss how at least one of these mechanisms might explain the recent success of B cell-depleting therapies for MS. While a better mechanistic understanding of the role of EBV infection and its immune control during MS pathogenesis is required and calls for the development of innovative experimental systems to test the proposed mechanisms, therapies targeting EBV-infected B cells are already starting to be explored in MS patients.

## 1. Introduction on EBV, Its Tumorigenesis and Its Immune Control

Epstein Barr virus (EBV) is a ubiquitous γ-herpesvirus that persistently infects more than 90% of the human adult population [1]. At the same time, it readily transforms human B cells into immortalized lymphoblastoid cell lines (LCLs) in culture and was identified as the first human tumor virus in Burkitt’s lymphoma [2,3]. Moreover, EBV is associated with additional lymphomas and epithelial carcinomas as well as smooth muscle tumors [4]. The majority of these tumor cells express latent EBV gene products that are not required for infectious viral particle production [5]. While Burkitt’s lymphoma expresses only EBV nuclear antigen 1 (EBNA1) as protein, Hodgkin’s lymphoma and nasopharyngeal carcinoma express EBNA1 and the two latent membrane proteins (LMP1 and 2). Finally, diffuse large B cell lymphomas (DLBCLs) in immune-suppressed patients often express six nuclear antigens (EBNA1, 2, 3A, 3B, 3C, and LP) and the LMPs. In addition, non-translated viral RNAs (EBERs and miRNAs) are expressed in all EBV-associated malignancies. Expression of all latent gene products can also be found in LCLs, and these transform B cells into potent antigen-presenting cells (APCs) [6]. These so-called viral latency patterns are, however, not exclusive to the EBV-associated tumors, but are also found in healthy EBV carriers [7]. All latent EBV proteins are expressed in naïve B cells. The three Hodgkin’s lymphoma-associated latent EBV proteins (EBNA1, LMP1, and LMP2) are found in germinal center B cells of healthy EBV carriers, and Burkitt’s lymphoma-associated sole EBNA1 expression is detected in homeostatically proliferating memory B cells [8]. EBV is thought to persist without latent EBV protein, but non-translated viral RNA expression in memory B cells as a long-lived lymphocyte reservoir [9]. Thus, the viral gene expression programs of EBV-associated malignancies are already established in healthy EBV carriers, but their development into malignancies is prevented by cytotoxic cell-mediated immune responses.

This becomes apparent during acquired or inherited immune suppression. For example, co-infection with the human immunodeficiency virus (HIV) leads to increased development of EBV-associated Burkitt’s lymphoma, Hodgkin’s lymphoma, and DLBCL [10]. In addition, primary immunodeficiencies that affect individual genes and render affected patients susceptible to EBV-associated pathologies point towards cytotoxic lymphocytes, primarily CD8^+^ T cells, as important pillars of EBV-specific immune control [11,12,13]. These identify the perforin/granzyme cytotoxic machinery as essential for EBV-specific immune control, while type I and II interferon responses are dispensable. Furthermore, they identify T cell receptor signaling and the co-receptors CD27, 4-1BB, SLAM family receptors, and NKG2D as important during EBV-specific immune control. Finally, additional primary immunodeficiencies indicate that the development and expansion of these cytotoxic lymphocytes ensures immune control of EBV in most individuals that are infected prior to two years of age [14]. This immune control needs to keep persistent infection by this important human tumor and immune modulatory virus under control for the rest of the host’s life.

## 2. Epidemiological Evidence for EBV’s Association with MS

As for many other autoimmune diseases, the etiology of multiple sclerosis (MS) is not entirely clear and does not completely rely on genetics as there are strong environmental factors influencing the individual susceptibility. Migration studies show a consistent pattern of individuals, who migrated from a high-risk MS area to an area with a low prevalence, acquiring the risk associated with the new host region. Interestingly, this tends to be only true if migration occurred during childhood before the age of 15. After this time window, individuals predominantly retain the MS risk from the area they migrated from [15,16]. These findings suggest that during childhood, environmental factors can modify the disease susceptibility and are relevant for the initiation of the disease, indicating that the first two decades of life are crucial for MS risk establishment.

For many years, an infectious etiology has been suspected. Epidemiological data supports the “hygiene hypothesis”, proposing the existence of infectious agents that increase the risk of MS if acquired in adolescence, but not if primary infection occurred during infancy [17,18]. Supporting this hypothesis, epidemiological findings indicate that MS prevalence is low in developing countries and tends to increase in regions with higher socioeconomic status and sanitation.

Interestingly, the time point of primary EBV infection is generally considered as a marker for childhood hygiene and has been linked to increased risk of MS development. Usually acquired in early childhood in developing countries, primary EBV infection is drastically postponed in developed areas, with a much lower seroprevalence in young adults [14,19]. For the latter, the risk of acquiring EBV dramatically increases and contraction will more likely be manifested as IM, whereas primary EBV infection during childhood is generally associated with low or no symptoms [20]. A study conducted in 2006 found that individuals with a history of infectious mononucleosis (IM), which is primary EBV infection with an overshooting CD8^+^ T cell lymphocytosis, carry a 3.2 times higher risk for developing MS compared to EBV-positive individuals who acquired the virus asymptomatically [21].

While several studies provide strong evidence for IM as an important risk factor for MS, the extremely high prevalence of EBV seropositivity in the population (95%) and the relatively low frequency of MS incidence pose a great challenge to prove the direct causality between EBV status and the risk for MS development without considering IM history. The first evidence for a positive association between EBV infection and the occurrence of MS came from a landmark longitudinal study. The collection of serum samples from more than 8 million active-duty US military personnel found no MS cases among individuals with an EBV seronegative status, while 100% of the individuals that developed MS seroconverted to an EBV-positive status prior to the onset of MS symptoms [22]. Additionally, two independent studies conducted in pediatric MS cases found increased EBV seropositivity in children with MS (89.6% and 83%), compared to healthy controls (72% and 42%) [23,24]. To date, these studies provided some of the strongest epidemiological evidence associating EBV infection to MS. In summary, EBV infection prior to the age of 15 might influence the risk of developing MS. This age dependency could result from altered immune responses upon virus encounter at adolescence and early adulthood. This will be discussed next.

## 3. Altered EBV-Specific Immune Responses in MS Patients

EBV has long been suspected to be involved in the pathogenesis of multiple autoimmune diseases. Several studies indicate that the humoral and cellular immune response against EBV as well as the regulation of viral persistence in the EBV-infected memory B cell pool are dysregulated in certain autoimmune diseases [25,26,27,28,29,30,31].

The first indications of an altered EBV-specific B cell response in MS came from several independent longitudinal studies. Analysis of serum samples, collected from a healthy adult population before the onset of MS disease, showed a significant increase of antibody titers to EBV nuclear antigens (particularly EBNA1) several years before the manifestation of the first MS symptoms [32,33,34]. This indicates that the risk of developing MS increases significantly with the levels of anti EBV antibody titers and the elevation of such antibodies is likely an early event in MS pathology. Further evidence for the involvement of a dysregulated EBV-specific immune response in MS came from the investigation of oligoclonal IgG antibodies in cerebrospinal fluid (CSF) of MS patients. Enrichment of clonally expanded B cells and accumulation of oligoclonal IgG in the CSF of MS patients are considered as immunological hallmarks of the disease [35,36]. Similar to the systemic increase of EBV-specific antibodies before MS onset, investigations found higher frequencies of CSF-derived EBNA1-specific IgG antibodies in MS patients [37,38,39,40]. In addition to EBNA1-specific antibodies, humoral responses to the lytic EBV antigens BRRF2 and BFRF3 were found to be elevated in MS patients [41]. Some cross-reactivity to self-proteins of these elevated EBV antigen-specific antibody responses was identified. Along these lines, some EBNA1-specific antibodies cross-react with anoctamin 2 [40], and BRRF2-specific antibodies bind mitochondrial proteins, while BFRF3-specific antibodies recognize septin-9 [42].

Evidence for the involvement of EBV-specific T cell responses in driving MS pathogenesis came from a study investigating the sequence similarity between T cell epitopes and self-peptides. Several virus-derived peptides, including a peptide derived from the DNA-polymerase protein of EBV (BALF5), were identified that had the ability to efficiently activate myelin basic protein (MBP)-specific CD4^+^ T cell clones [43]. An increased frequency of EBNA1-specific Th1-polarized CD4^+^ T cells, with more diversified EBNA1 recognition, has been found in MS patients compared to healthy EBV seropositive controls [28]. In accordance with this, a subsequent study showed increased CD4^+^ T cell proliferation of PBMCs from MS patients stimulated with EBNA1 peptides but not with peptides from influenza or HCMV. Such MS patient-derived CD4^+^ T cells showed a significantly higher recognition of self-myelin antigen compared with other autoantigens [44]. Several studies also found that EBV-specific CD8^+^ T cell responses are significantly higher in MS patients compared to healthy controls and patients with other inflammatory neurological diseases [45,46]. By using HLA class I pentamers, it was shown that CD8^+^ T cell responses to EBV lytic antigens peaked during active disease and are lower during the inactive phases of MS, demonstrating that changes in the immune response to EBV are associated with the different phases of MS [47]. Interestingly, in many studies, no difference was found in EBV viral loads between the MS and control groups [28,45], indicating that although MS patients are able to efficiently control EBV infection, at least systemically, their CD4^+^ as well as their CD8^+^ EBV-specific T cell responses are dysregulated and might rather promote disease pathology. Thus, both EBV-specific antibody and T cell responses are elevated in MS patients, and some of these cross-react with autoantigens.

## 4. Evidence for Decreased Immune Control of EBV in MS Patients and Genetic MS Predisposition

Genes within the human leukocyte antigen (HLA) complex have long been known to play a crucial part in the development of MS and other autoimmune diseases. Genome-wide association studies identified the HLA allele DRB1*15:01 (HLA-DR15) as the strongest genetic risk factor of MS [48]. Interestingly, symptomatic primary EBV infection, IM, has been found to synergize with this main genetic risk factor HLA-DR15, leading to a 7-fold increase in MS risk [49]. The underlying mechanism of this synergistic effect is, however, largely unknown. Efforts to unravel this interaction have so far been hampered by the lack of an adequate model to study this interaction in vivo.

By using a humanized mouse model of EBV infection, we could recently confirm a possible synergistic effect between EBV and the HLA complex HLA-DR15. Engraftment of immune-deficient mice with either HLA-DR15^+^ or HLA-DR15^−^ human immune system components showed that animals carrying the HLA-DR15 haplotype suffered from higher viral loads compared to those engrafted with HLA-DR15^−^ cells [50]. Interestingly, these elevated viral titers in HLA-DR15^+^ animals were accompanied by an increased expansion of CD8^+^ T cells and higher T cell activation, similar to the clinical manifestation of IM symptoms [50], implicating a poor MHC class II-mediated immune control of EBV infection in HLA-DR15 carriers (Figure 1). Additionally, EBV-reactive CD4^+^ T cells from HLA-DR15^+^ animals showed reactivity against MBP, one of the major autoantigens in MS [50]. Similarly, autoantigen RASGRP2-specific CD4^+^ T cells restricted by HLA-DRB5*0101, a MHC class II molecule in linkage disequilibrium with HLA-DR15 and belonging to the MS-associated HLA haplotype, cross-react with a peptide of the EBV large tegument protein deneddylase BPLF1 [51]. This supports previous data suggesting molecular mimicry as the mechanism for the induction of CNS autoreactive T cells during EBV infection (Figure 1). In summary, these findings suggest that EBV infection in the context of the main genetic risk factor for MS, HLA-DR15, leads to a reduction of EBV-specific immune control, thereby supporting the priming of hyperreactive and cross-reactive T cells.

Regarding the interaction between EBV-specific B cell immune response and HLA-DR15, it was found that HLA-DR15^+^ individuals had elevated antibody activity to the EBV antigen EBNA1 compared to HLA-DR15-negative carriers. Furthermore, in HLA-DR15 carriers, less EBNA1 reactivity was required to increase the risk for MS development [52]. In a follow-up study, specific serological responses against EBNA1 epitopes and their association with HLA-DR15 were investigated. MS patients showed an overall elevation of antibody reactivity against several different domains of EBNA1 compared to matched controls. However, antigenicity varied greatly between different domains. The strongest association with increased MS risk came from the EBNA1 domain (amino acid 385–420) [53]. Interestingly, this fragment contains a pentapeptide that shares homology with the heat shock peptide αB-crystallin. Upon EBV infection, αB-crystallin can be upregulated in peripheral B cells. This EBV-induced expression leads to the priming of pro-inflammatory T cells, which could trigger myelin-directed autoimmunity [54]. Indeed, αB-crystallin is often found in MS lesions and has been suggested to serve as an immunodominant myelin antigen in the CNS of MS-affected individuals [54]. Altogether, EBV infection in the context of HLA-DR15 could lead to the activation and expansion of virus-specific T and B cells that cross-react to myelin antigens and initiate the pathology of MS.

## 5. Role of B Cells and Their Antigen Presentation during MS

However, it remains unclear why preferentially EBV infection and no other pathogens would explore genetic susceptibility for MS development. One possibility is that EBV converts B cells into potent APCs that then could stimulate autoreactive, possibly EBV cross-reactive CD4^+^ T cells [55]. Along these lines EBNA2 has been shown to augment transcription of MS risk loci, resulting in increased LCL proliferation [56,57]. Some of these affect tumor necrosis factor (TNF) receptor signaling [58] that is also engaged by LMP1, which augments antigen presentation of LCLs [59]. The recent success of B cell-depleting therapies in MS suggests that indeed, B cells might fulfill an important function as APCs in this autoimmune disease [60,61,62,63]. These successful therapies target CD20 and deplete naïve and memory B cells but leave plasma cells and for some time antibody levels untouched in the treated patients. In contrast, Atacicept, which targets naïve B cells and plasma cells, exacerbated MS and therefore its clinical trials were halted [64]. This suggests that activated and memory B cells promote MS and clinical improvement is already achieved shortly after their depletion.

Some of these activated B cells might be EBV transformed and localize to the CNS for autoimmune T cell restimulation (Figure 1), especially during progressive disease. Along these lines, B cell follicles and especially meningeal B cell follicles are associated with severe disease progression [65,66]. These might be supported by follicular helper T cells and B cells with NF-κB activation as identified by single-cell RNA sequencing in the cerebrospinal fluid [67,68]. In these follicles, some studies have found EBV-infected B cells [69,70,71], while others have not [72,73]. In addition, viral transcripts were so far not detected in the cerebrospinal fluid [68]. A propensity of EBV-infected B cells to home to the CNS has been observed in other disease settings. These include primary CNS lymphomas that occur in HIV-infected individuals [74,75]. Those are 100% EBV-associated and occur after loss of EBV-specific CD4^+^ T cells when systemic immune control of EBV is still fairly intact. A second clinical setting in which EBV-transformed B cells home to the brain and are associated with clonal EBV-specific T cell expansions in the CNS are some patients that suffer from neurological symptoms after immune check-point blockade of PD-1 [76]. Thus, B cell infiltration and follicle formation in the CNS are associated with progressive MS disease, but it remains unclear if these include significant numbers of EBV-infected B cells, which, however, in other clinical settings demonstrate a propensity to home to the brain.

Nevertheless, under the assumption that such an EBV-transformed B cell reservoir exists in the CNS or periphery and is insufficiently controlled but eliminated by B cell-depleting therapies, clinical trials were started to eradicate these cells with EBV-specific T cells [77,78,79]. For this purpose, MS patient-derived EBV-specific T cells were expanded with an adenovirus encoding EBNA1, LMP1, and LMP2 epitopes. These were then transferred back into so far 10 MS patients. Sustained clinical responses that correlate with the EBV reactivity of the original T cell product were observed for up to 3 years [79]. Albeit in few patients, these findings suggest that EBV transformation might contribute to the activated memory B cell compartment that promotes MS and is depleted by CD20-specific antibody therapies, such as with Rituximab, Ocrelizumab, and Ofatumumab. However, future research is needed to localize this EBV-infected B cell reservoir that might drive autoimmune T cells by antigen presentation.

## 6. Conclusions and Outlook

EBV infection and altered immune responses during IM have now been strongly associated with an increased risk of developing MS [55,80,81,82]. It is less clear, however, if this results from an uncontrolled reservoir of B cells, activated by EBV infection, that stimulates autoreactive T cells, possibly even in the CNS, or if the infection primes cross-reactive T cell responses that recognize both EBV and myelin antigens. Some evidence for both has been reported, but recent clinical data on the success of B cell-depleting therapies and adoptive transfer of EBV-specific T cells would favor the former hypothesis. Such dysregulation of EBV infection and its specific immune control is associated with increased MS risk and can be found in MS patients. This argues in favor of strengthening the EBV-specific immune control, possibly even by vaccinating susceptible individuals long before MS onset. Along these lines, adolescents that are still EBV seronegative and therefore have an increased risk of developing IM upon primary infection [14] might benefit from such a vaccination, similar to vaccination against human papillomavirus (HPV) at this age [83]. Different recombinant viral vectors, EBV-derived virus-like particles, and recombinant viral proteins are currently being explored for vaccination against EBV-associated tumors [84,85,86,87,88,89,90,91,92,93], and it will be interesting if these could prevent IM and the associated risk of developing MS, or even treat MS. Before targeting MS with EBV-specific vaccination, however, a better understanding of the mechanisms by which EBV exploits genetic susceptibility to increase MS risk is needed.

## Figures and Tables

**Figure 1 microorganisms-09-02191-f001:**
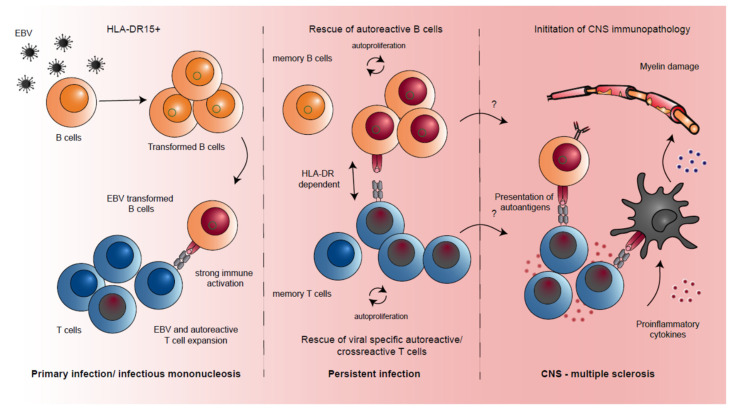
Potential interaction mechanisms between EBV infection and HLA-DR15 for the development of MS. HLA-DR15^+^ carriers show a reduced EBV-specific immune control, with higher viral loads and increased T cell activation. The strong immune activation during primary EBV infection could allow the activation and expansion of autoreactive T cells by insufficiently controlled EBV-transformed B cells (autoreactivity depicted as red nuclei). Accumulating EBV-transformed B cells could activate cross-reactive T cells, favoring their survival and expansion, in an HLA-DR-dependent manner. By a yet unknown mechanism, autoproliferative lymphocytes might home to the CNS, where possibly EBV-transformed B cells could act as APCs for the cross-reactive T cells that then activate myeloid cells with proinflammatory cytokines to start the immunopathology seen in MS with myelin damage.

## Data Availability

Not applicable.

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
