# Peer review of "Epstein Barr Virus Exploits Genetic Susceptibility to Increase Multiple Sclerosis Risk"

_microorganisms, 2021, doi:10.3390/microorganisms9112191_

Round 1

Reviewer 1 Report

Dr. Laderach and Munz submit a review of EBV and genetic susceptibility to MS.  The relation of EBV and MS is a large topic with an extensive literature.  They do a good job of reviewing the evidence associating EBV and MS, and have a good closing discussion of the role of B cells in MS and possible ways that EBV could be involved.  The references include several from this year.  I do not have any suggestions for revisions.  There were a few typographical errors, as listed below. 

page 4, line 159, should be “to synergize”. 

Page 4, line 172 should be MBP. 

Page 6, line 224 “achieved”

Author Response

We thank both reviewers for their encouraging comments. We have now corrected all minor typos and changed the color code in the figure to distinguish more clearly B and T cells, as well as autoreactive and infected lymphocytes. We outline the changes below and by underlining in the revised manuscript.

Minor comments:

I do not have any suggestions for revisions.  There were a few typographical errors, as listed below.

Page 4, line 159, should be “to synergize”.

This has been corrected.

Page 4, line 172 should be MBP.

This has been corrected.

Page 6, line 224 “achieved”

This has been corrected.

Reviewer 2 Report

This review article focuses on the evidence linking active infection with Epstein Barr virus (EBV) to genetic susceptibility for the development of multiple sclerosis (MS). The authors present the evidence that symptomatic EBV infection is linked to increased risk for development of MS. They also provide evidence that MS patients exhibit decreased immune control of EBV in a manner dependent on HLA-DR15, the strongest genetic risk factor for MS. They provide evidence for a model in which individuals expressing HLA-DR15 exhibit reduced EBV immune control, and that these altered responses might be linked to the generation of autoreactive T cell responses. They propose that B cells play a role by presenting autoantigens to cross-reactive T cells, which is in line with the therapeutic activities of anti-CD20 antibodies against MS. They also propose that the development of effective EBV vaccines may provide a means to prevent or possibly even treat MS.

General comments: The review is well-written, focused, timely, and provides a nice perspective on the link between EBV infection, MHC class II alleles, and susceptibility to MS, a topic that remains relatively controversial. The review is well-balanced and contains appropriate references. A nice figure is provided. I only have a few minor suggestions for improvement.

Minor comments:

1. The figure is nice but I had difficulties distinguishing between the different cell types. B and T cells appear to be different shades of blue that are hard to distinguish, so I would suggest using different colors for B and T cells. Are red nuclei used to indicate both autoreactive T cells and insufficiently controlled EBV transformed B cells? If so, maybe differently colored nuclei could be used for B cells vs T cells.

2. Minor typos:

Line 35: replace “3c” by “3C”

Line 72: replace “suggest, that” by “suggest that”

Line 128: replace “were” by “was”

Line 160: replace “are” by “is”

Line 175: replace “hablotype” by “haplotype”

Line 224: replace “achieve” by “achieved”

Author Response

We thank both reviewers for their encouraging comments. We have now corrected all minor typos and changed the color code in the figure to distinguish more clearly B and T cells, as well as autoreactive and infected lymphocytes. We outline the changes below and by underlining in the revised manuscript.

General comments: The review is well-written, focused, timely, and provides a nice perspective on the link between EBV infection, MHC class II alleles, and susceptibility to MS, a topic that remains relatively controversial. The review is well-balanced and contains appropriate references. A nice figure is provided. I only have a few minor suggestions for improvement.

Minor comments:

  1. The figure is nice but I had difficulties distinguishing between the different cell types. B and T cells appear to be different shades of blue that are hard to distinguish, so I would suggest using different colors for B and T cells. Are red nuclei used to indicate both autoreactive T cells and insufficiently controlled EBV transformed B cells? If so, maybe differently colored nuclei could be used for B cells vs T cells.

We have now changed the colors to distinguish better B and T cells, as well as autoreactive and infected lymphocytes.

  1. Minor typos:

Line 35: replace “3c” by “3C”

This has been corrected.

Line 72: replace “suggest, that” by “suggest that”

The comma has been removed.

Line 128: replace “were” by “was”

This has been corrected.

Line 160: replace “are” by “is”

This has been corrected.

Line 175: replace “hablotype” by “haplotype”

This has been corrected.

Line 224: replace “achieve” by “achieved”

This has been corrected.